# SiCoDEA: A Simple, Fast and Complete App for Analyzing the Effect of Individual Drugs and Their Combinations

**DOI:** 10.3390/biom12070904

**Published:** 2022-06-28

**Authors:** Giulio Spinozzi, Valentina Tini, Alessio Ferrari, Ilaria Gionfriddo, Roberta Ranieri, Francesca Milano, Sara Pierangeli, Serena Donnini, Federica Mezzasoma, Serenella Silvestri, Brunangelo Falini, Maria Paola Martelli

**Affiliations:** Department of Medicine and Surgery, Section of Hematology and Clinical Immunology, University of Perugia, 06129 Perugia, Italy; valentina.tini@studenti.unipg.it (V.T.); alessio.ferrari86@gmail.com (A.F.); ilaria.gionfriddo@unipg.it (I.G.); roberta.ranieri@unipg.it (R.R.); francesca.milano@unipg.it (F.M.); sara7pierangeli@libero.it (S.P.); sere.donnini@gmail.com (S.D.); federicam8@hotmail.it (F.M.); silvestri.serenell@gmail.com (S.S.); brunangelo.falini@unipg.it (B.F.)

**Keywords:** drug screenings, bioinformatics, IC_50_, genomics, isobologram, combination index, synergic, antagonist, additive, automatic report

## Abstract

The administration of combinations of drugs is a method widely used in the treatment of different pathologies as it can lead to an increase in the therapeutic effect and a reduction in the dose compared to the administration of single drugs. For these reasons, it is of interest to study combinations of drugs and to determine whether a specific combination has a synergistic, antagonistic or additive effect. Various mathematical models have been developed, which use different methods to evaluate the synergy of a combination of drugs. We have developed an open access and easy to use app that allows different models to be explored and the most fitting to be chosen for the specific experimental data: SiCoDEA (Single and Combined Drug Effect Analysis). Despite the existence of other tools for drug combination analysis, SiCoDEA remains the most complete and flexible since it offers options such as outlier removal or the ability to choose between different models for analysis. SiCoDEA is an easy to use tool for analyzing drug combination data and to have a view of the various steps and offer different results based on the model chosen.

## 1. Introduction

When in 1965 Emil Frei designed the first combinatorial regimen for acute leukemia [1], it became clear that remissions obtained with therapies based on single drugs were only temporary, and that the clinical responses achieved were more durable when more agents were combined.

Over time, pursuing this approach, cancers that had previously been fatal such as acute lymphocytic leukemia, diffuse large B-cell lymphoma, Hodgkin’s lymphoma and testicular cancer have become largely curable [2].

These days effective chemotherapies mostly involve combinations of two or more drugs, allowing the targeting of tumor heterogeneity, feedback loops, dependencies and synthetic lethality, and the selective rise of therapy-resistant tumor clones.

In the last two decades, a new concept of cancer therapy, the targeted therapy, has emerged giving rise to several classes of cancer drugs that are designed to precisely block specific pathways that are relatively selective to the cells of distinct cancer types, inhibiting their growth or promoting their differentiation, or death sparing healthy tissues. These new targeted therapies that include kinase inhibitors, receptor inhibitors and immunotherapies give promising results in combination with standard chemotherapeutic regimens [3,4], and equally growing is the depth of molecular characterization in cancer.

With countless possibilities of combinations, dosing, scheduling and repurposing, and finer targeting due to the deeper disease characterization, the number of possible clinical trials vastly exceeds the number of patients [5].

If we take into account that just 7% of anti-cancer drugs undergoing clinical trials prove to be effective [6], while oncology trials are, together with cardiovascular trials, the most expensive [7], it becomes even more clear that technological advances are urgently needed to make the design of combination treatments, necessary to improve clinical outcomes for most patients, truly fruitful.

For these reasons, it is of interest to study combinations of drugs and to determine whether a specific combination has a synergistic, antagonistic or additive effect, i.e., greater, less than or equal to the effect expected by the sum of the individual drugs.

The fundamental step is the definition of a Combination Index (CI) that allows evaluation of the effect of the two drugs used separately with respect to the combination. The CI represents a value that indicates the distance of the observed response from the expected response and will indicate synergy if CI < 1, antagonism if CI > 1 and additivity if CI = 1. For this purpose, an *effect-based strategy* or a *dose–effect-based strategy* can be used [8], two different approaches that compare the observed effect of the combination with the expected effect under the assumption of non-interaction predicted by a reference model. In the first case, a direct comparison is made between the effect of the individual drugs and the effect of the drugs on the combination; in the second case the calculation of the dose–effect curves for the individual drugs is used to calculate their expected values. In this second case, therefore, a further important step is the choice of the model to calculate the dose–effect curve.

Each of these methods for calculating the CI has advantages and disadvantages based on the situation and the data being analyzed, and for this reason it is of interest to provide a choice. Creating a model for this purpose and calculating its parameters, however, requires a certain level of mathematical and programming knowledge or the use of commercial software.

There are already several tools that analyze the interaction between drugs, and the most famous or complete are CompuSyn [9], SynergyFinder Plus [10] and DDCV [11]. However, we have observed that, for an analysis that is as precise and flexible as possible, something is missing from each of these tools (Table 1). CompuSyn is a paid software that only works on Windows platforms, it also does not offer many options to choose from and only allows analysis using the median-effect model. DDCV is another shiny app that can be found on the web that allows the analysis of drug combinations. It uses only the median-effect model for the calculation of the CI, without the possibility of choosing otherwise. Another fairly complete app that allows the analysis of drug combinations is SynergyFinder Plus, but, although it allows you to choose between different models for the calculation of the CI, it does not allow you to choose between different models of dose–response curves in the web version. In addition, none of these tools offer an analysis of the outliers with a variable filter threshold to refine the model.

For this purpose, therefore, we have developed an open access and easy to use app that allows different models to be explored and the most fitting to be chosen for the specific experimental data: SiCoDEA (Single and Combined Drug Effect Analysis).

## 2. Materials and Methods

### 2.1. SiCoDEA Strategy

The models implemented within SiCoDEA for the calculation of CI are the effect-based strategy and dose–effect-based strategy.

The effect-based strategy includes:*Response additivity model.* Additionally known as the linear interaction effect, the response additivity model [12] consists of comparing the effect of the combination and the effect obtained by adding that of the individual drugs at the same dose. It is therefore possible to obtain a CI from the ratio
(1)$$CI_{Additivity}=\frac{E_{A}+E_{B}}{E_{AB}},$$
where *E_A_* is the effect of drug A alone, *E_B_* is the effect of drug B alone and *E_AB_* is the effect of drugs A and B in combination.*Highest single agent (HSA) model*. The highest single agent model (HSA) or Gaddum’s non-interaction model [13] assumes that the expected effect of the combination is equal to the highest effect of the individual drug at the same dose as it has in the combination. Thus, a synergistic combination should produce an additional beneficial effect compared to what individual drugs alone can achieve. The CI is given by the difference between the effect of the combination at a given dose and the highest effect of one of the single drugs at that same dose
(2)$$CI_{HSA}=\frac{\max\left(E_{A},\;E_{B}\right)}{E_{AB}}.$$*Bliss independence model*. The Bliss independence model [14] assumes a stochastic process in which two drugs produce their effect independently. Therefore, the expected effect of the combination can be calculated as the probability of two independent events: $E_{A}+E_{B}-E_{A}E_{B}$ where $0\leq E_{A}\leq 1$ and $0\leq E_{B}\leq 1$. The CI will be
(3)$$CI_{Bliss}=\frac{E_{A}+E_{B}-E_{A}E_{B}}{E_{AB}}.$$The dose–effect-based strategy includes:*Loewe additivity model*. The principle on which the Loewe additivity model [15,16,17] is based is that to calculate the CI it is necessary to compare the doses of the drugs in combination with the doses of the individual drugs necessary to achieve the same effect. In this way, if the dose required for a single drug is lower than that in combination, we will have an antagonistic effect between the two drugs, if, instead, it is higher, the effect will be synergistic. The CI will be calculated as
(4)$$CI_{Loewe}=\frac{a}{A}+\frac{b}{B},$$
where 𝑎 is the dose of drug A in the combination, 𝐴 is the equivalent dose, i.e., the dose of drug A needed to achieve the same effect of the combination, 𝑏 is the dose of drug B in the combination and 𝐵 is the equivalent dose, i.e., the dose of drug B needed to achieve the same effect of the combination.*Zero Interaction Potency (ZIP) model*. The Zero Interaction Potency (ZIP) model [18,19] combines the Loewe model and the Bliss model together if, in combination, the two dose–effect curves do not change. It then uses the same calculation of two independent events as the Bliss model, but using the values calculated through the dose–effect curve as in the Loewe model. The CI will be
(5)$$CI_{ZIP}=\frac{EE_{A}+EE_{B}-EE_{A}EE_{B}}{E_{AB}},$$
where 𝐸𝐸_𝐴_ is the expected effect of drug A, and 𝐸𝐸_𝐵_ is the expected effect of drug B calculated based on the dose–effect curve.

In the case of dose–effect-based approaches, the choice of the model for the calculation of the dose–response curve is equally important. Although, in fact, there are models that are more used than others, it is important to evaluate case by case, based on the data under examination, which model is best suited. In SiCoDEA, the following 5 models are available to choose*Chou–Talalay Method* [20] (*median-effect*), which is the most commonly used model based on the median-effect equation, derived from the mass action law principle
(6)$$D=D_{m}\cdot {\left(\frac{f_{a}}{1-f_{a}}\right)}^{\frac{1}{m}}.$$
(7)$$f_{a}=\frac{1}{1+{\left(\frac{D_{m}}{D}\right)}^{m}}.$$
where 𝐷 is the dose of interest, 𝐷_𝑚_ is the median-effect dose, 𝑓_𝑎_ is the fraction affected and 𝑚 is the slope.

Another widely used model is the *log-logistic one*, which can use two, three or four parameters: with two parameters, the minimum is set equal to zero and the maximum equal to one; with three parameters only one of the two is kept fixed (either the maximum or the minimum); finally, with four parameters there is no fixed value, but all four are calculated within the model (maximum, minimum, median-effect dose, 𝐷_𝑚_, and slope, 𝑚). The following are the formulas for the four models*Log-logistic with four parameters (log-logistic).*



(8)
$$D=D_{m}\cdot {\left(\frac{max-min}{f_{a}-min}-1\right)}^{\frac{1}{m}}.$$





(9)
$$f_{a}=min+\frac{max-min}{1+{\left(\frac{D}{D_{m}}\right)}^{m}}.$$




*Log-logistic with three parameters (minimum equal to zero, log-logistic[0]).*




(10)
$$D=D_{m}\cdot {\left(\frac{max-f_{a}}{f_{a}}\right)}^{\frac{1}{m}}.$$





(11)
$$f_{a}=\frac{max}{1+{\left(\frac{D}{D_{m}}\right)}^{m}}.$$




*Log-logistic with three parameters (maximum equal to one, log-logistic[1]).*




(12)
$$D=D_{m}\cdot {\left(\frac{1-min}{f_{a}-min}-1\right)}^{\frac{1}{m}}.$$





(13)
$$f_{a}=min+\frac{1-min}{1+{\left(\frac{D}{D_{m}}\right)}^{m}}.$$




*Log-logistic with two parameters (log-logistic[01]).*




(14)
$$D=D_{m}\cdot {\left(\frac{1-f_{a}}{f_{a}}\right)}^{\frac{1}{m}}.$$





(15)
$$f_{a}=\frac{1}{1+{\left(\frac{D}{D_{m}}\right)}^{m}}.$$



Outliers’ analysis:

Here, a Grubbs’ test [21] is used to check whether there is an outlier in the data that is significantly different from the other values. For this purpose, a G score is calculated, obtained as $G=\frac{O-mean}{SD}$ where O represents the value of the presumed outlier, mean is the average of all values and SD the standard deviation. Based on this score, the *p*-value is then calculated.

SiCoDEA (available at https://sicodea.shinyapps.io/shiny/, accessed on 20 April 2022) is developed through a *shiny* interactive and easy to use interface (R based); it provides both the simple calculation of the IC_50_ for different drugs and the calculation of the CI with the display of the respective plots. The utilized R packages are “shiny” [22], “shinyjs” [23], “plyr” [24], “car” [25], “drc” [26], “ggplot2” [27], “tidyr” [28], “gplots” [29], “outliers” [30], “scales” [31], “rlist” [32], “dplyr” [33].

### 2.2. SiCoDEA Validation

The validation of our novel SiCoDEA open-source app was performed with drug combinations on acute myeloid leukemia (AML) cell lines (OCI-AML3 and OCI-AML2), cultured as recommended by cell line providers and previously described [34,35,36]. For analysis, AML cells were exposed to drugs or controls either as single or combination treatment.

The first step was the optimization of drug responsive curves for the drugs used as single agents. Cells were seeded in 384-well plate at a concentration of 4 × 10^5^ cells/mL in a volume of 22.5 µL/well and treated in triplicate with either vehicle (negative control) or 7 log-scale concentrations of each drug using the automized D300e Digital Dispenser (Tecan, Männedorf, Switzerland). Treatment with 100 µM Etoposide (a topoisomerase II inhibitor with known wide anti-tumor activity) was used as positive control. After either 48 or 72 h, cell proliferation was assessed using the cell metabolism independent CyQUANT Direct Cell Proliferation Assay (Life Technologies, New York, NY, USA). Cells were incubated for 4 h with CyQUANT reagent, and then fluorescence signal of drug-treated and vehicle-treated samples were measured using a Spark Microplate Reader (Tecan). Data were normalized assuming 100% cell proliferation to vehicle-treated control and 0% proliferation to 100 µM Etoposide treatment.

Once drug–response curves had been optimized, we proceeded to the drug combinatorial treatment. Again, cells were seeded in 384-well plates and treated in triplicate with an 8 × 8 matrix of log-scale drug concentrations ranging from 0 to the maximum effective dose for each drug. The IC_50_ of each drug was in the middle of the 7 drug doses. CyQUANT reagent was added 48 or 72 h later, and data were analyzed to assess the CI.

As example of SiCoDEA application to drug combinatorial treatment analysis, here we tested the combination of ABT-199 (Venetoclax, Catalog No. S8048) and HHT (Homoharringtonine, also named Omacetaxine mepesuccinate, Catalog No. S9015) on the model of OCI-AML3, carrying nucleophosmin (NPM1) gene mutation [34]. Both drugs were purchased from Selleck Chemicals (Houston, TX, USA). Drugs were prepared as 10 mM stocks in 100% dimethyl sulfoxide (DMSO) and added to culture media at the final concentration for drug assay. Here, OCI-AML3 cells were treated for 48 h with HHT at dose range: 5 × 10^−10^–1 × 10^−07^ M, and ABT-199, at dose range: 2 × 10^−09^–5 × 10^−06^ M.

## 3. Results

### 3.1. Implementation and Description of SiCoDEA Functions

A detailed step by step description of the SiCoDEA app used with a combination of a generic Drug A and Drug B is available in Appendix A.

The app is divided into three tabs, each dedicated to a different analysis.

The first tab (Figure 1) is dedicated to single drug analysis and has the advantage of being able to analyze many drugs in a single file. It is sufficient to enter one drug per line, both in the dose file and in the response file. Once the data have been loaded, the single drug analysis consists of evaluating the trend of the dose–response curves to correctly choose the parameters relating to the normalization method and *p*-value threshold for the outlier test. It is possible to choose between two different normalization methods, one based on the maximum value and the other on the value calculated at drug concentrations equal to zero. Before the analysis, a test is also carried out to check for the presence of outliers in the data, which can compromise the calculation of the model parameters. Our app gives the possibility to observe the dose–response curves to evaluate the possible presence of outliers and remove them based on the *p*-value threshold.

At this step the choice of the model for the calculation of the dose–response curve is equally important. Although, in fact, there are models that are more used than others, it is important to evaluate case by case, based on the data under examination, which model is best suited. In our app, the plot shows the curves for all five models, as well as the line corresponding to the IC_50_ for each model. Based on the curves and the calculated R^2^, the model that best fits the data can be chosen (Figure 2).

In the second tab (Figure 3) it is possible to make a comparison between different samples, such as the different cell lines on which the same drugs are administered at the same doses. The input files are identical to those of the previous tab, with the difference being that up to four files with drug response data can be loaded. Thus, t is possible to choose between the different models, also based on what has been seen in the previous tab, and show the curve corresponding to each drug. It can be represented by one to four curves, based on the number of samples you want to analyze and compare.

Finally, in the third tab (Figure 4) we have the analysis relating to the combination of drugs. Here we have both the visualization relating to the single curves for the five models of the dose–response curve with the calculation of the R^2^, and the plots for the combination, with the representation of the CI. It is possible to choose between both the five models of the combination index and between the five dose–response curves, in the case of dose–effect-based methods.

For the chosen options, a plot is created that shows the trend of the CI for the different drug combinations and, consequently, whether it is synergistic, antagonistic, or additive.

Finally, it is possible to export the results in single png files or in a summary report in pdf.

### 3.2. Application of SiCoDEA

We are currently using SiCoDEA for drug data analysis of AML cells exposed to different drugs combinations on study in our laboratory. In particular, we are studying the effect of HHT (Homoharringtonine, also named Omacetaxine mepesuccinate) and ABT-199 (Venetoclax) on models of acute myeloid leukemia (AML), specifically AML with *NPM1* mutation, which is the most frequent AML in adult patients, accounting for about one-third of all cases [37]. Venetoclax is a small molecule drug that directly and selectively inhibits the B-cell leukemia/lymphoma 2 (Bcl-2) anti-apoptotic protein [38], highly expressed in many hematologic malignancies, including AML. Omacetaxine mepesuccinate is a synthetic form of the plant cephalotaxine alkaloid homoharringtonine (HHT) that is derived from the bark and leaves of various Cephalotaxus species. HHT binds to the A-site cleft of ribosomes preventing the initial elongation step of protein synthesis and leading to a transient but profound inhibition of the synthesis of proteins, especially those with a short half-life such as the myeloid cell leukemia 1 (Mcl-1) anti-apoptotic protein, often upregulated in leukemic cells [39]. Following the inhibition of Bcl-2, the activation of Mcl-1 is a known mechanism of resistance that the cell develops to inhibit apoptosis.

Here, we present representative data derived by the application of SiCoDEA to drug effect analysis, generated by treating AML cell lines with HHT and ABT-199 (hereinafter indicated as ABT) and using the CyQUANT Direct Cell Proliferation Assay to evaluate the anti-proliferative effect, as described in the Section 2.

First, we treated OCI-AML2 (not harboring *NPM1* mutation) versus OCI-AML3 (carrying *NPM1* mutation) with HHT for either 48 or 72 h and compared the drug anti-proliferative effects. In the reported experiment, observing the curves generated for OCI-AML2 at 72 h of treatment (Figure 5a), we see the presence of some outliers for low dosages of the drug, which compromise the quality of the dose–response curves and further analysis. Indeed, it is evident that the curves of the various models are significantly influenced by the presence of these outliers and, consequently, they are less consistent with the observed data. The calculated R^2^ also reflects this trend, as they report values below 0.9, while the goodness of fit of a model is evaluated by the proximity of this value to 1. Applying the outlier removal function of SiCoDEA, we could appreciate a clear improvement in the quality of the dose–response curves (Figure 5b). Indeed, the models adapt better to the observed data and the R^2^ values rise to values above 0.9.

Curves obtained on OCI-AML3 are displayed in Figure 5c.

For comparison analysis (Figure 5d), we chose one model whose R^2^ values are or approach the best for each experimental condition. In this case, the best model for each corresponded to the log-logistic with three parameters model (*Log-logistic[1]*).

This analysis showed that the HHT IC_50_ at 72 h for OCI-AML3 and OCI-AML2 are similar and in the range of 7–9 nM, in keeping with what was previously reported [40].

In order to predict the synergistic effects of the HHT/ABT combination in AML with the *NPM1* mutation to be translated into the clinics, we tested SiCoDEA with the OCI-AML3 cell line treated as described in the Section 2. By normalizing the starting data and calculating the average for the various replicates, we first obtained a heatmap showing the inhibition values for each drug combination (Figure 6a). We then generated the curves for each drug according to the five different models available (Figure 6b,c). Specifically, in the case of the HHT drug, the best model (and therefore with a higher R^2^) was the logistic with three parameters (*log-logistic[1]*, yellow line) (Figure 6b), with a calculated IC_50_ of 1.55 × 10^−08^ M, while in the case of the ABT drug, the best model was the log-logistic with four parameters (*log-logistic*, purple line), with a calculated IC_50_ of 7.345 × 10^−06^ M (Figure 6c).

The next step was the choice of the model that best suits the two drugs under examination. Since the model must be the same for the two drugs when evaluating a combination, we chose the two-parameter log-logistic model (*log-logistic[01]*), which was the one that on average gave better results in both the curves.

For the chosen options, a plot was then created that shows the trend of the CI for the different drug combinations and, consequently, whether it is synergetic, antagonistic, or additive (Figure 7).

Combination efficacy analysis shows a combination index (CI value) of less than 0.1 (Figure 7, blue dots) at multiple combination concentrations, indicating that ABT in combination with HHT has a synergistic anti-proliferative effect in the OCI-AML3 cell line. In particular, a strong synergy was observed in correspondence with combinations of clinically relevant concentrations of either HHT (between 1.7 × 10^−08^ and 4.1 × 10^−08^ M, being the pharmacological concentration of about 15–30 nM) or ABT (between 5.8 × 10^−08^ and 3.2 × 10^−07^ M, being the pharmacological concentration of about 100–200 nM) (Figure 7). Strikingly, in in vivo preclinical experiments in patient-derived xenograft (PDX) murine models of AML with *NPM1* mutation, the combinatorial HHT/ABT treatment with doses equivalent to the pharmacological concentrations, confirmed a strong synergistic anti-leukemic activity and gave a significant survival advantage compared with single drug or vehicle-treated animals [41]. These preclinical findings greatly contributed to the approval of a phase 1 clinical trial entitled “A Phase I Study to Evaluate Safety and Preliminary Efficacy of Omacetaxine Mepesuccinate (Synribo) Combined with Venetoclax (Venclyxto) in Patients with Relapsed/Refractory Acute Myeloid Leukemia with Nucleophosmin (NPM1) Gene Mutation” (EudraCT n. 2019-001821-29), active in recruiting patients at our Hematology Clinical Department [41].

## 4. Discussion

The purpose of SiCoDEA is to provide an easy to use tool for analyzing drug combination data, to have a view of the various steps and to offer different results based on the model chosen. An important prerequisite in analyzing drug combinations is in fact the dose–response curve calculated for individual drugs. Many of the existing tools, from the famous CompuSyn to the most recent SynergyFinder Plus, involve the use of a single model in the calculation of the dose–response curve, but we have observed that there is no universally better model than the others: different data require different models. For this reason, we decided to implement in SiCoDEA five different models for calculating the dose–response curve, as well as five different models for calculating the CI (Table 1). SiCoDEA allows you to view the plots of the individual drugs with the curves of the different models taken into consideration and evaluate which one best fits the data and thus has the best R^2^ value. A table showing all the R^2^ values for the five different models is automatically created with the curve plot. This is certainly an advantage since it ensures the best adherence of the model to real data and therefore increases the quality of the analysis.

Furthermore, as described above, since the presence of outliers can compromise the results of the analyses, SiCoDEA includes the possibility of choosing the *p*-value threshold to use for the outlier calculation, based on the Grubbs’ test [21]. By changing the threshold, you can immediately observe the consequences on the model to choose the right value to obtain the most reliable results.

Another type of analysis that may prove useful and that we intend to implement in the future is the one that involves the three-drug combination.

Other advantages include that it is open source and works on different platforms. Moreover, it allows you to download a final report, and for each type of analysis, it is possible to export the results in single png files or in a summary report in pdf.

In conclusion, SiCoDEA is an open-source app and among the most complete. It is developed through a shiny interactive and easy to use interface (R based, as with our RNA-Seq app, ARPIR [42,43]) and it allows users to perform three different types of analysis:Single drug analysis. Obtain the dose–response curve for different drugs by uploading a single file. For each drug, curves are displayed for all five models with relative R^2^ and IC_50_.Drug comparison. Compare the effect of the same drug on different samples, up to a maximum of four, by choosing the most fitting model from the five options.Drug combination analysis. Perform a combination analysis in two steps: first, visualizing the dose–response curves for the five models in the two drugs considered; second, based on the R^2^, choosing the best model to be adopted for the dose–response curve and for the CI. Results are displayed in an isobologram plot and in a heatmap.

Data input can be easily integrated into Laboratory Information Management Systems (LIMS), which support analysis plate readings, such as adLIMS [44].

SiCoDEA can be a useful tool, next to the already existing ones, for drug combinatorial treatment analysis as it introduces different mathematical models allowing accurate fitting of data. It is a flexible app that allows you to better adapt the analysis parameters based on the data and can be further improved by adding the three-drug combination analysis option.

## Figures and Tables

**Figure 1 biomolecules-12-00904-f001:**
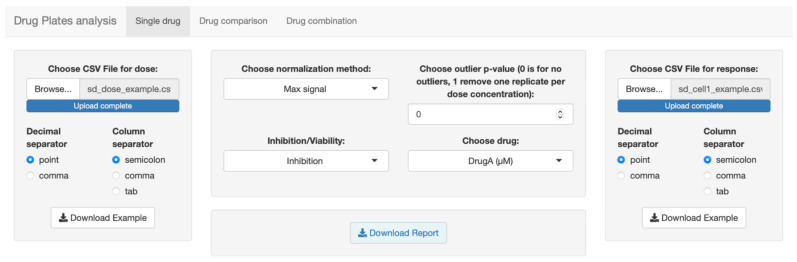
Screenshot of the first tab, dedicated to single drug analysis.

**Figure 2 biomolecules-12-00904-f002:**
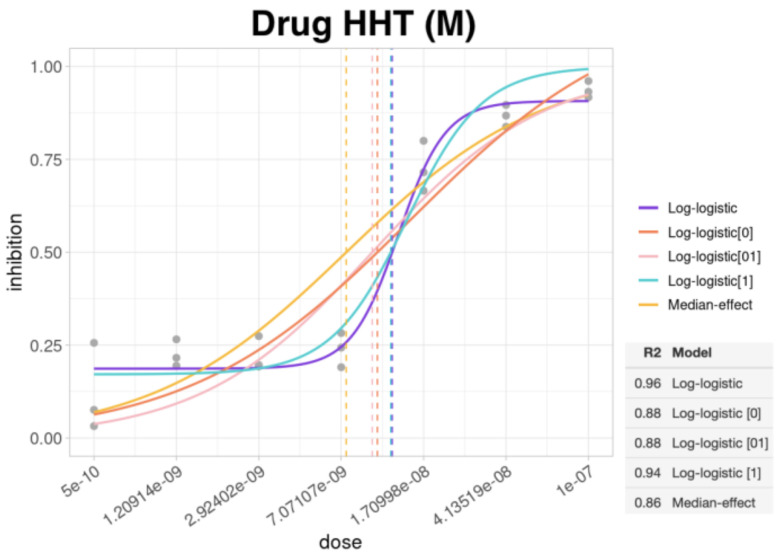
Dose–response curves applying five different models to the same data. The HHT drug concentration is shown in the x-axis and the proportion of cells (OCI-AML2) that have undergone inhibition in the y-axis. The lower right table shows the R^2^ value for each. In this case, the best models with the highest R^2^ and therefore best suited to the data are the *log-logistic* (R^2^ 0.96) (purple line) and the *log-logistic[1]* (R^2^ 0.94) (light blue line).

**Figure 3 biomolecules-12-00904-f003:**
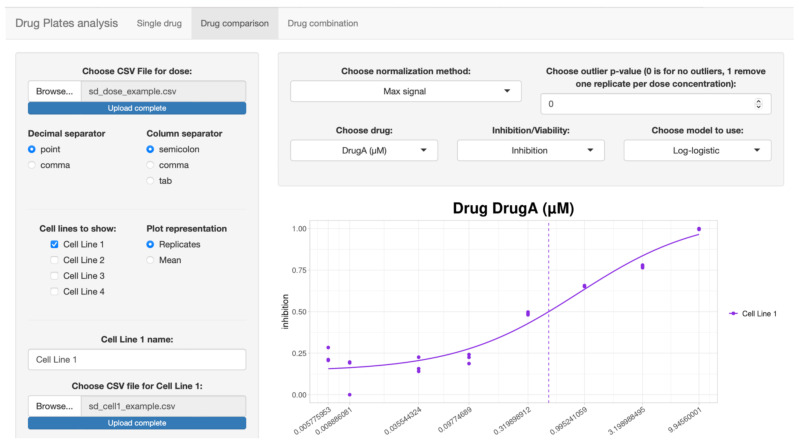
Screenshot of the second tab, dedicated to drug comparison.

**Figure 4 biomolecules-12-00904-f004:**
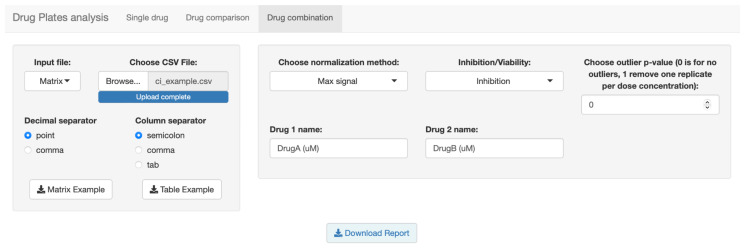
Screenshot of the third tab, dedicated to drug combination.

**Figure 5 biomolecules-12-00904-f005:**
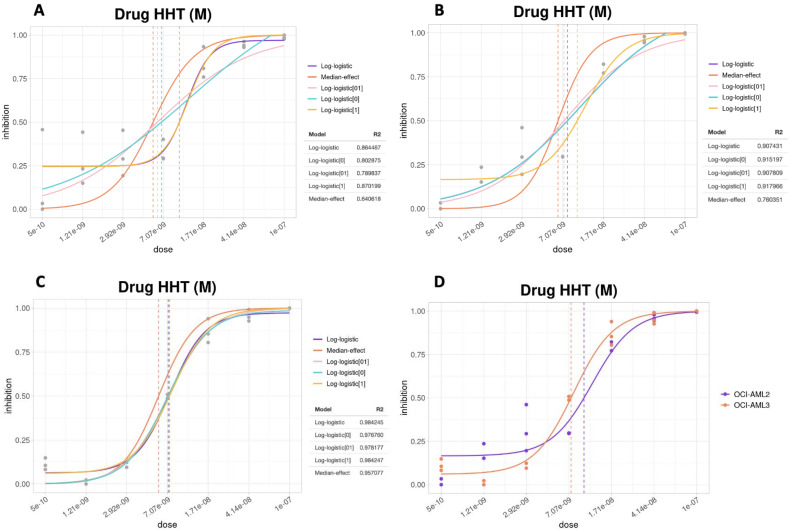
Dose–response curves for HHT on OCI-AML2 and OCI-AML3 at 72 h. (**A**,**B**) Curves obtained with OCI-AML2 according to the different models and data before (**A**) and after (**B**) outliers’ removal. (**C**) Curves obtained with OCI-AML3 according to the different models. (**D**) Comparison between OCI-AML2 and OCI-AML3 cell lines treated with HHT. The concentration of the drug is shown on the x-axis and the proportion of inhibition is shown on the y-axis.

**Figure 6 biomolecules-12-00904-f006:**
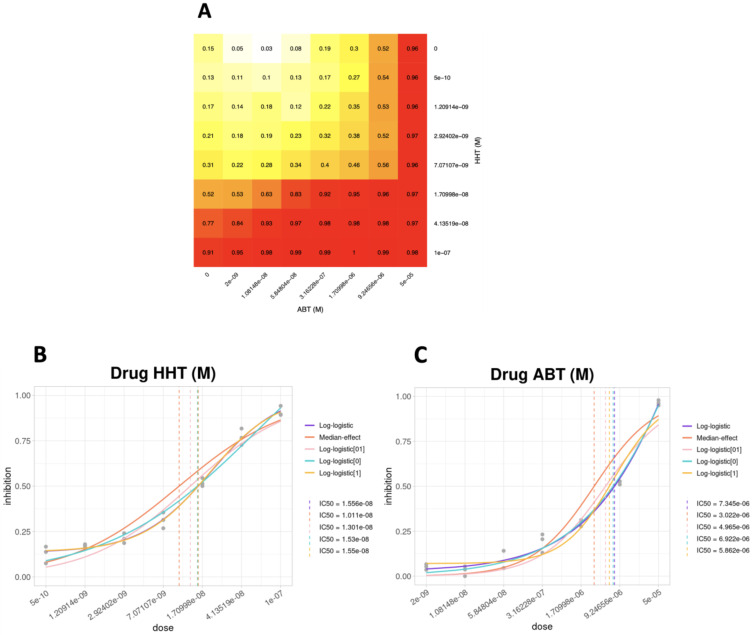
Combinatorial HHT/ABT drug treatment analysis in OCI-AML3. (**A**) Heatmap of normalized inhibition levels. (**B**,**C**) Dose–response curves according to the five different models available in SiCoDEA, for OCI-AML3 cell line treated with HHT drug (**B**) and ABT drug (**C**) for 48h. The drug concentration is shown in the x-axis and the proportion of cells that have undergone inhibition in the y-axis.

**Figure 7 biomolecules-12-00904-f007:**
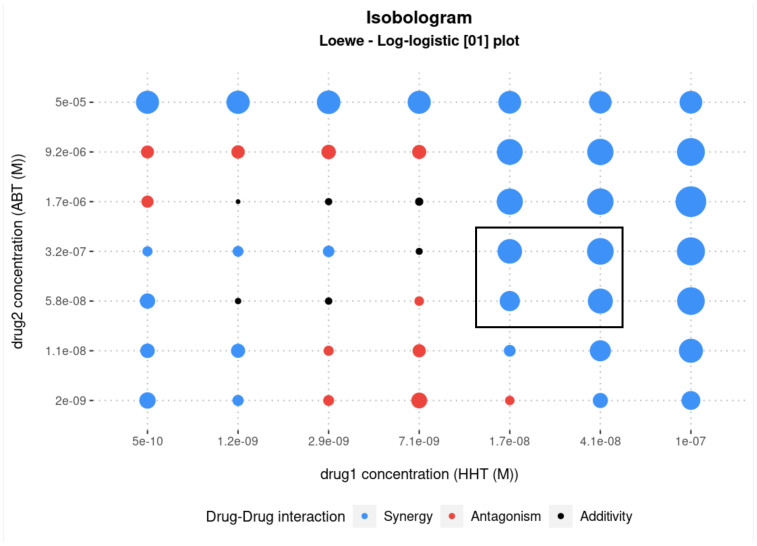
Combination index (CI) values calculated for OCI-AML3 cell line treated with the drug combination HHT/ABT for 48 h. Here, the model used for the calculation of the dose–response curve is the log-logistic with two parameters (*log-logistic[01]*) from the data shown in Figure 6, and for the calculation of the combination index of the *Loewe* model. The higher the size of the circle is, the higher the CI power. The black square highlights the synergistic effect obtained with clinically relevant drug concentrations of HHT (x-axis) and ABT (y-axis).

**Table 1 biomolecules-12-00904-t001:** Options available in the various tools for drug combination analysis.

	CI Models	Drug–Response Models	Open Source	Report	Single Drug Analysis	Customizable Outlier Analysis	Platform	3 Drugs Analysis
SynergyFinder Plus	4	1	🗸	🗸	🗸		Win/Mac/Linux	🗸
CompuSyn	1	1			🗸		Win	
DDCV	1	1	🗸	🗸			Win/Mac/Linux	
SiCoDEA	5	5	🗸	🗸	🗸	🗸	Win/Mac/Linux	

## Data Availability

The datasets supporting the conclusions of this article are included within the article (and its Additional Appendix A). The software is available in the GitHub repository: https://github.com/giuliospinozzi/SiCoDEA (accessed on 20 April 2022).

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
