# Peer review of "SiCoDEA: A Simple, Fast and Complete App for Analyzing the Effect of Individual Drugs and Their Combinations"

_biomolecules, 2022, doi:10.3390/biom12070904_

Round 1

Reviewer 1 Report

Giulio et al. are reporting a software in this manuscript. SiCoDEA is a R-based  software to conduct data analysis of administration of individual drugs or combinations of drugs. The combination index (CI) is comprehensively evaluated using different effect-based strategies. Plots shows the software interfaces and example dose-response curves.

The reviewer would say that the description for the process of software development is clear and solid. The software can potentially benefit researchers to automate the data analysis procedure. 

The reviewer would also point out that no scientific question is answered in this manuscript. This manuscript is purely presenting a new software. Developing new tools and software is also appreciated to the community. But please make sure it is within the journal's scope.

The reviewer would suggest moving the section of describing effect-based strategy from Introduction to Materials and Methods, as these strategies are essentially "Methods". 

Author Response

Giulio et al. are reporting a software in this manuscript. SiCoDEA is a R-based software to conduct data analysis of administration of individual drugs or combinations of drugs. The combination index (CI) is comprehensively evaluated using different effect-based strategies. Plots shows the software interfaces and example dose-response curves.

The reviewer would say that the description for the process of software development is clear and solid. The software can potentially benefit researchers to automate the data analysis procedure.

The reviewer would also point out that no scientific question is answered in this manuscript. This manuscript is purely presenting a new software. Developing new tools and software is also appreciated to the community. But please make sure it is within the journal's scope.

Reply: We thank the Reviewer for the comment. Although here our main aim was to describe the development and functions of SiCoDEA, we agree that presenting data obtained by the use of this software would enrich the manuscript. Therefore, we have added details on experimental conditions used to validate our software and images related to data obtained with SiCoDEA with a novel drug combination including ABT-199 (venetoclax) and HHT (homoharringtonine, also named omacetaxine mepesuccinate), that we have studied in our laboratory in the setting of acute myeloid leukemia, and lead to the clinics in a phase 1 clinical trial active in recruiting patients at our Hematology Clinical Department (Title: “A Phase I Study To Evaluate Safety And Preliminary Efficacy Of Omacetaxine Mepesuccinate (Synribo) Combined With Venetoclax (Venclyxto) In Patients With Relapsed/Refractory Acute Myeloid Leukemia With Nucleophosmin (NPM1) Gene Mutation”, EudraCT n. 2019-001821-29). Lines from 163 to 193, and 253 to 347.

The reviewer would suggest moving the section of describing effect-based strategy from Introduction to Materials and Methods, as these strategies are essentially "Methods".

Reply: We agree with the Reviewer and better organized the Material and Methods section, moving the part describing the “SiCoDEA strategy” from Introduction to Materials and Methods: lines from 90 to 161.

Reviewer 2 Report

Spinozzi et al. developed an open access and easy-to-use application for single and combined drug effect analysis. It provides a user-friendly web application, in which multiple models of analyses are available for user to choose and compare. The analyses include outlier removal, dose-response curve fitting, multiple drug comparison and drug combination index calculation. I can foresee that this web application will be very useful for experimental data analysis and facilitate the design of combination treatments.

However, there is no new insights provided by this manuscript in terms of drug discovery. The results section only introduced the available functions and analyses of this web server, lacking an exact example of application. I recommend the authors adding at least one example in the manuscript that uses this web application to analyze experimental data and provides new insights of the combination effects for some drugs.

In addition to my major concern, I have the following minor comments:

(1)   At line 83, there is a typo in “where 0 ?A 1e0 ?B 1.” Please correct it.

(2)   I think Equation (7) is not correct. fa should be (1+(Dm/D)^m)^(-1). I recommend the authors double check their code to see if they have implemented the correct equation.

(3)   The outlier removal strategy, as one of the highlighted functions of this application, needs to be described in detail in the Method section.

Author Response

Spinozzi et al. developed an open access and easy-to-use application for single and combined drug effect analysis. It provides a user-friendly web application, in which multiple models of analyses are available for user to choose and compare. The analyses include outlier removal, dose-response curve fitting, multiple drug comparison and drug combination index calculation. I can foresee that this web application will be very useful for experimental data analysis and facilitate the design of combination treatments.

However, there is no new insights provided by this manuscript in terms of drug discovery. The results section only introduced the available functions and analyses of this web server, lacking an exact example of application. I recommend the authors adding at least one example in the manuscript that uses this web application to analyze experimental data and provides new insights of the combination effects for some drugs.

Reply: We thank the Reviewer for the comment. Although here our main aim was to describe the development and functions of SiCoDEA, we agree that presenting data obtained by the use of this software would enrich the manuscript. Therefore, we have added details on experimental conditions used to validate our software and images related to data obtained with SiCoDEA with a novel drug combination including ABT-199 (venetoclax) and HHT (homoharringtonine, also named omacetaxine mepesuccinate), that we have studied in our laboratory in the setting of acute myeloid leukemia, and lead to the clinics in a phase 1 clinical trial active in recruiting patients at our Hematology Clinical Department (Title: “A Phase I Study To Evaluate Safety And Preliminary Efficacy Of Omacetaxine Mepesuccinate (Synribo) Combined With Venetoclax (Venclyxto) In Patients With Relapsed/Refractory Acute Myeloid Leukemia With Nucleophosmin (NPM1) Gene Mutation”, EudraCT n. 2019-001821-29). Lines from 163 to 193, and 253 to 347.

In addition to my major concern, I have the following minor comments:

(1)   At line 83, there is a typo in “where 0 ≤ ?A ≤ 1e0 ≤ ?B ≤ 1.” Please correct it.

Reply: Thanks for the suggestion, we edited the text and fixed the typos: line 110.

(2)   I think Equation (7) is not correct. fa should be (1+(Dm/D)^m)^(-1). I recommend the authors double check their code to see if they have implemented the correct equation.

Reply: We thank the reviewer for the careful revision. Indeed, there was a typo in the text, and equation (7) now has been corrected. We have checked also the software and it is ok (it is confirmed that it was only a typo in the text).

(3)   The outlier removal strategy, as one of the highlighted functions of this application, needs to be described in detail in the Method section.

Reply: Thanks for the suggestion, we added a detailed description in Materials and Methods section: lines from 150 to 155.

Reviewer 3 Report

This manuscript, SiCoDEA: a simple, fast and complete app for analyzing the effect of individual drugs and their combinations, described a novel and useful app to analyze the drug effect of single or combined usage, as stated in the title. This is a good and useful tool. But in my opinion, this manuscript was not well written and a major revision is needed.

Comments:

1.     The introduction section is too long, almost half of the paper. The authors should concentrated on the results or discussion sections to tell the novelty of the app. And at the same time, do not write the results like a direction of the app.

2.     The interface of the app is very simple like three tables. I believe the authors can design it more beautiful.

3.     There are some mistakes such as IC50 without subscript 50 and R2 without superscript 2.

Author Response

This manuscript, SiCoDEA: a simple, fast and complete app for analyzing the effect of individual drugs and their combinations, described a novel and useful app to analyze the drug effect of single or combined usage, as stated in the title. This is a good and useful tool. But in my opinion, this manuscript was not well written and a major revision is needed.

Comments:

1.     The introduction section is too long, almost half of the paper. The authors should concentrate on the results or discussion sections to tell the novelty of the app. And at the same time, do not write the results like a direction of the app.

Answer: We agree with the Reviewer and edited the text moving from the Introduction to Materials and Methods section the paragraphs describing SiCoDEA strategy, that are more properly related to methods, as suggested also by another Reviewer: lines from 90 to 161. Moreover, we have expanded the discussion section underlying the novelty of SiCoDEA: lines from 349 to 393

2.     The interface of the app is very simple like three tables. I believe the authors can design it more beautiful.

Reply: We appreciate the suggestion of the Reviewer. We decided to use shiny for its simplicity and compatibility. This allows us to release an application that is simple to install and very light, even for computers with few resources. This ease of use is documented both in the guide and in the video tutorial that we have included on YouTube, also attached in the text. On the other hand, we realize that we do not have an aesthetically appealing application. We have decided to place greater emphasis on compatibility and the standard. Shiny is now the most widely used platform for tools of this type, natively integrating the entire reporting and internal data management system. In any case, we are committed to giving more weight to aesthetics with the next updates, if the Shiny API allows it.

3.     There are some mistakes such as IC50 without subscript 50 and R2 without superscript 2. –

Reply: Thanks for the suggestion, we have fixed all the mistakes.

Reviewer 4 Report

The article by Spinozzi et al. describes SiCoDEA - an open access pplication that analyzes the individual effect of drugs as well as the effect of their combinations. The topic is of high interest for the preclinical drug analysis, the paper is well written and rich in information, the language is clear and concise. I recommend the publication of the manuscript in Biomolecules, after clarifying the following:
-    Please provide an abbreviation for the specific models of the Dose-Effect-Based approaches according to the ones in the plots;
-    In figures 1A and 1B the same model - log-logistic (purple line) - seems to provide the best fit. However, from the description of the figure 1 and also Discussions section, there should be different models (four and three parameters respectively).
-    For greater impact, examples of real drugs can be given instead of drug A and drug B.
-    To be fair, please add SiCoDEA in table 1. I would also add the “combination of 3 drugs” feature provided by Synergy Finder Plus as an advantage;
-    I recommend organizing a section for a detailed description of the advantages given by SiCoDEA.

Author Response

The article by Spinozzi et al. describes SiCoDEA - an open access application that analyzes the individual effect of drugs as well as the effect of their combinations. The topic is of high interest for the preclinical drug analysis, the paper is well written and rich in information, the language is clear and concise. I recommend the publication of the manuscript in Biomolecules, after clarifying the following:
-    Please provide an abbreviation for the specific models of the Dose-Effect-Based  approaches according to the ones in the plots;

Reply: Thanks for the suggestion, we have added all the abbreviations

-    In figures 1A and 1B the same model - log-logistic (purple line) - seems to provide the best fit. However, from the description of the figure 1 and also Discussions section, there should be different models (four and three parameters respectively).

Reply: We thank the Reviewer for the comment. We apologize for being unclear in the text. Due to extensive revision of the data presentation, previous figures are now replaced and the name of the models are reported in both the Figures and the text.

-    For greater impact, examples of real drugs can be given instead of drug A and drug B.

Reply: We thank the Reviewer for the comment. Although here our main aim was to describe the development and functions of SiCoDEA, we agree that presenting data obtained by the use of this software would enrich the manuscript. Therefore, we have added details on experimental conditions used to validate our software and images related to data obtained with SiCoDEA with a novel drug combination including ABT-199 (venetoclax) and HHT (homoharringtonine, also named omacetaxine mepesuccinate), that we have studied in our laboratory in the setting of acute myeloid leukemia, and lead to the clinics in a phase 1 clinical trial active in recruiting patients at our Hematology Clinical Department (Title: “A Phase I Study To Evaluate Safety And Preliminary Efficacy Of Omacetaxine Mepesuccinate (Synribo) Combined With Venetoclax (Venclyxto) In Patients With Relapsed/Refractory Acute Myeloid Leukemia With Nucleophosmin (NPM1) Gene Mutation”, EudraCT n. 2019-001821-29). Lines from 163 to 193, and 253 to 347.

-    To be fair, please add SiCoDEA in table 1. I would also add the “combination of 3 drugs” feature provided by Synergy Finder Plus as an advantage

Reply: We thank the Reviewer for the suggestion and agree we mistakenly skipped this feature that is now reported as an advantage for Synergy Finder Plus in Table 1. However, we know the importance of having it in our tool, so we are also planning to include it in the near future, as soon as we have also set up a pilot experiment with three drugs. Mathematical models already exist.

-    I recommend organizing a section for a detailed description of the advantages given by SiCoDEA.

Reply: We thank the Reviewer for the suggestion. We have inserted in Table 1 the characteristics of SiCoDEA and expanded the discussion specifying better the advantages of the tool we developed.

Round 2

Reviewer 2 Report

The authors have addressed all my concerns. Most importantly, they have added a detailed example of how to use their web apllication to analyze the experimental data, and presented the validation of their software using a novel drug combination of ABT-199 and HHT, which strengthens the significance of content. I support the publication of this manuscript.